# Pediatric Anesthesia Providers’ Perspective on the Real-Life Implementation of the Philips Visual Patient Avatar: A Qualitative Study

**DOI:** 10.3390/children10121841

**Published:** 2023-11-24

**Authors:** Justyna Lunkiewicz, Daniel Fries, Petar Milovanovic, Christoph B. Noethiger, David W. Tscholl, Greta Gasciauskaite

**Affiliations:** Institute of Anesthesiology, University and University Hospital Zurich, Raemistrasse 100, 8091 Zurich, Switzerland; justyna.lunkiewicz@usz.ch (J.L.); daniel.fries@usz.ch (D.F.); petar.milovanovic@usz.ch (P.M.); christoph.noethiger@usz.ch (C.B.N.); greta.gasciauskaite@usz.ch (G.G.)

**Keywords:** pediatric anesthesia, situation awareness, user-centered design, patient monitoring, Visual Patient Avatar, qualitative research

## Abstract

The Philips Visual Patient Avatar represents an alternative method of patient monitoring that, according to computer-based simulation studies, enhances diagnostic accuracy and confidence and reduces workload. After its clinical integration, we assessed pediatric anesthesia providers’ perspectives on this technology. This is a single-center qualitative study, conducted at the University Hospital Zurich using in-depth individual interviews. We aimed to identify the advantages and limitations of the Visual Patient Avatar in pediatric anesthesia and to assess children’s and parents’ reactions from caregivers’ perspectives. Thematic analysis was used to identify the dominant themes. Fourteen members of the institution’s pediatric anesthesia team were interviewed. The most prevalent themes were children’s positive reactions towards the Visual Patient Avatar (92.9%) and enhanced speed in problem identification (71.4%). Additionally, 50% of participants reported finding the Visual Patient Avatar useful for diverting children’s attention during anesthesia induction, and 50% suggested that its vital sign thresholds should be adaptable for different age groups. The study revealed that the Visual Patient Avatar was recognized as a tool in pediatric anesthesia, enabling prompt identification of underlying issues and receiving positive feedback from both children and parents. The most commonly voiced wish for improvement in the study was the ability to customize the Visual Patient Avatar’s thresholds for different age groups.

## 1. Introduction

Advancements in medical technology have significantly enhanced patient monitoring in the fields of perioperative and intensive care medicine [1,2]. Patient monitoring to date predominantly relies on the single-sensor–single-indicator principle—a technology-centered approach that involves measuring specific parameters and displaying them individually as separate numbers and waves [3]. However, this model does not fully exploit the potential of human sensory perception [4]. The Philips Visual Patient Avatar—a situation awareness-oriented visualization technology for patient monitoring (Figure 1)—was specifically developed according to human-centered design principles to make optimal use of human users’ perceptive abilities. It uses colors, shapes, and animations to convey information about vital signs [4] and displays multiple vital signs derived from various sensors in a single indicator—i.e., a patient avatar [5,6]. Based on cognitive psychology and neuroscience principles [7,8,9,10], such a user-centered visual language is more effectively perceived, comprehended, and interpreted by humans, leading to an enhanced level of situation awareness [4,11]. Situation awareness is based on gathering information from diverse environmental sources, comprehending its significance, and employing it to anticipate future developments. It entails developing and maintaining a dynamic awareness of the situation and associated risks within an activity [5]. While the principles of situation awareness initially emerged within the aviation context, they have also found application in intensive care medicine [12] and anesthesia [13], where the effective management of dynamic situations is paramount. Notably, approximately 80% of treatment errors in intensive care units and intraoperative settings can be attributed to insufficient situation awareness among healthcare providers [14,15]. The positive impacts of situation awareness enhancing technology—e.g., the Visual Patient Avatar—have been demonstrated in numerous simulation studies [4,11,16]. This visualization technology has enabled clinicians in a high-fidelity simulation study to identify the underlying medical issue with a higher probability than conventional monitoring. Additionally, it has been shown to improve diagnostic confidence and alleviate perceived workload in computer-based studies [4]. Studies showing the impact of the real-world implementation of the Visual Patient technology are ongoing and not yet published, as the technology is newly introduced into clinical practice. Some other user-centered patient monitoring technologies such as AlertWatch:OR, AlertWatch:OB, AlertWatch:AC (AlertWatch, Inc., Ann Arbor, MI, USA) [17] have demonstrated positive results in real-world clinical practice. AlertWatch:OR showed statistically significant improvements in several process measures, including crystalloid administration, tidal volume, hypotension [18], and intraoperative glycemic management [19]. However, implementing data displays that aid anesthesia providers in detection, diagnosis, and treatment is challenging [20]. Previous user perception studies following simulation studies have revealed some concerns raised by participants, such as a lack of accurate quantitative data presentation, overloaded avatar presentation, or some design issues, e.g., visualization of very high pulse and respiratory rates, which are sometimes difficult to distinguish [21]. One of the design considerations is the limited understanding of medical cognition in anesthesia [20]. However, this can be addressed by building on the research of Sowb and Loeb [22], as well as the work of Roth [23], who conducted cognitive analyses of specific tasks. One of the cognitive analyses included the task of managing the ventilation system. The analysis included interviews with clinicians in and outside the operating room, observations of clinicians in their practice, analysis of critical events, literature reviews, and an engineering-based analysis of a mechanical system. The information gathered from these various techniques was used to create a problem-solving model for clinician management of the ventilation system and to develop monitoring devices that could improve clinician diagnosis of various pulmonary conditions. Further insight can be gained from the research of Patel and colleagues [24], who used knowledge-elicitation techniques to uncover the mental model of the cardiovascular system held by physicians. There are a variety of knowledge-assessment methods, as discussed by Cooke [25], that can be applied to the field of anesthesia. Augmenting these methods with eye-tracking technology can help address the challenge of implicit knowledge. By combining these approaches, we can gain a deeper understanding of the anesthesiologist’s mental model, which in turn can inform the design of displays in this context [20].

While the functionality and user perceptions of the Visual Patient Avatar have been examined in multiple simulation studies [11,16], there is currently a lack of information regarding the evaluation of this technology in clinical practice, as it was introduced into clinical settings only in the spring of 2023.

The objective of this study was to capture and descriptively analyze the perspectives of pediatric anesthesia personnel, as well as indirectly those of pediatric patients and their parents or legal guardians, regarding the use of the Visual Patient Avatar in clinical routine. Specifically, we asked participants about their perceived positive aspects of the Philips Visual Patient Avatar in pediatric anesthesia, areas for improvement, and the perceived reactions of children and their parents or guardians to the technology.

## 2. Materials and Methods

### 2.1. Approval and Consent

The study protocol was assessed by the Cantonal Ethics Committee of the Canton of Zurich, Switzerland, who issued a declaration of non-jurisdiction (Business Management System for Ethics Committees number Req-2021-00756). Nevertheless, each participant gave written informed consent for their data to be used for research purposes. Participation was voluntary and without any compensation.

### 2.2. Study Design

We conducted a researcher-initiated, single-center, qualitative descriptive study investigating anesthesia providers’ perceptions of the Visual Patient Avatar in pediatric anesthesia. The study involved in-depth interviews with anesthesia providers, including staff anesthesiologists, residents, and nurses. We initially contacted all pediatric anesthesia providers (21) at the study center. All participants were required to have at least 6 months of pediatric anesthesia experience prior to the study and to have worked with the Philips Visual Patient Avatar on a regular clinical basis for at least 1 month. During the interviews, field notes were collected. The study was performed at the Institute of Anesthesiology, University Hospital Zurich, Switzerland, over two consecutive weeks in April 2023.

### 2.3. The Visual Patient Avatar in the Study Center

A situation-awareness-oriented, user-centered visualization technology for patient monitoring—Visual Patient Avatar—was introduced in the operating theatres of the University Hospital Zurich in March 2023. The University Hospital Zurich, where the technology was developed, is the pioneering center for implementing the Visual Patient Avatar in clinical practice. During the initial phase of the technology’s implementation, lectures explaining the concept of the Visual Patient Avatar and training sessions were held to provide information on the use of the technology. An educational video and a quick guide were made available on the Intranet of the study center (Appendix A). As in adult anesthesia, the Visual Patient Avatar is routinely used in pediatric anesthesia in the study center as a split-screen feature, which displays a combination of conventional patient monitoring and Visual Patient Avatar side-by-side (Figure 2). In the study center, three monitor profiles—neonatal, pediatric, and adult—are used. These profiles differ by their defined acoustic alarm limits and Visual Patient thresholds and are selected according to the patient category.

### 2.4. Reflexivity

Reflexivity is a set of ongoing, collaborative, and multifaceted processes through which researchers self-consciously criticize, evaluate, and assess how their subjectivity and context affect the research processes [26]. With this in mind, we provide information about different types of reflexivity as they relate to our research. The authors of this study are researchers involved in the development of the Visual Patient Avatar technology, which is the main theme of personal reflexivity in this current research, as they are motivated to evaluate the feedback and address the concerns of the users. The main issue of interpersonal reflexivity in this context is the fact that the study participants work at the center where the Visual Patient technology and other situation awareness-based user-centered technologies have been developed, which may influence their understanding and awareness of this issue. In terms of methodological reflexivity, the qualitative approach was chosen to allow for a comprehensive analysis of the perceptions of the new tool and to collect as many comprehensive answers to the open-ended questions as possible, as this is the first study to analyze users’ perceptions after the real-life implementation of the technology. Other methodological approaches, such as mixed methods, may be beneficial in future research. Regarding contextual reflexivity, as the increasing complexity of patient monitoring systems creates an ever-increasing density of information to be perceived and understood, it is paramount to support care providers in this context. One solution may be to provide user-centered visualizations of vital signs data.

### 2.5. Data Collection

We collected data through semi-structured individual interviews, using an interview guide to provide a general framework for discussion (Appendix A). The interview guide was designed by the study authors and was divided into two parts: open-ended questions and demographic data. The process of creating and validating the interview guide included several key steps. First, we established clear research objectives to identify the positive aspects and areas for improvement of the Philips Visual Patient Avatar in pediatric anesthesia and to understand how both children and parents respond to the Philips Visual Patient Avatar from a caregiver perspective. We then developed open-ended questions aligned with our research objectives. These questions were intentionally broad to encourage participants to freely share their experiences. In addition, demographic questions were designed to be non-specific to ensure participant anonymity. Demographic information was collected at the beginning of each interview. This included age, gender, role (staff anesthesiologist, resident, nurse), and the number of years participants had practiced anesthesia. The final question was whether the respondent was familiar with the educational materials on the Visual Patient Avatar (Appendix A). Input was obtained from anesthesia team members not involved in the study to assess content validity and refine the questionnaire. A pilot study with three anesthesia team members helped identify issues related to question wording, comprehension, and response options. Revisions were made based on their feedback.

We implemented a comprehensive study design to proactively mitigate potential sources of bias. To minimize nonresponse bias [27], we ensured the clarity and brevity of the questionnaire. To reduce the impact of self-selection bias [28], we conducted presentations on the Philips Visual Patient Avatar supported by scientifically proven simulation studies, as real-life studies are still ongoing. To address information bias [29], we carefully designed the survey instrument, following a structured process described above to establish its validity and reliability.

### 2.6. Data Analysis

#### 2.6.1. Open-Ended Questions

We translated responses to open-ended questions from German to English using DeepL (DeepL GmbH, Cologne, Germany)—an online translator. The translated answers are provided in Appendix A.

We used the thematic analysis six-phase approach to identify the topics that dominated the participants’ responses in each open-ended question separately. The six-step approach includes these key steps: familiarizing yourself with the data, generating initial codes, searching for themes, reviewing themes, defining and naming themes, and writing the report [30]. After completing the data collection and carefully reviewing the gathered information, the research team engaged in a discussion regarding their overall impressions and potential thematic ideas based on the collected responses. First, three team members, DF, DWT and GG, analyzed the participants’ answers independently; then, they made a final decision regarding the main themes provided by the participants in a joint discussion. We provide the number of respondents who mentioned each theme during the interview and the corresponding percentages.

#### 2.6.2. Demographic Data

We conducted the demographic data analysis using Microsoft Excel (Microsoft Corp., Redmond, WA, USA) and Prism 9 (GraphPad Software, Boston, MA, USA). We present the demographic data as numbers and their percentage distribution or as medians and interquartile ranges.

## 3. Results

### 3.1. Participant Characteristics

Of the 21 anesthesia care providers on the institution’s pediatric anesthesia team, 14 (66.7%) were available for the interviews due to limited availability because of clinical responsibilities, lack of experience in pediatric anesthesia, lack of work experience with the Visual Patient technology, and willingness to participate in an interview, which determined the sample size for the study. The distribution of staff anesthesiologists, residents, and nurses was approximately equal in number. The least experienced care provider had 2 years of anesthesia practice, whereas the most experienced had 27 years. All participants had at least six months of pediatric anesthesia experience. Detailed participant characteristics are shown in Table 1.

### 3.2. The Positive Impact of Visual Patient Avatar on Pediatric Anesthesia from Anesthesia Providers’ Perspective

Two themes were identified to summarize the positive aspects of the Visual Patient Avatar in the pediatric setting: (1) faster identification of problems through visual information, resulting in increased situation awareness, and (2) usefulness of the tool for distracting children and reducing stress during anesthesia induction.

#### 3.2.1. Faster Identification of Problems through Visual Information, Resulting in Increased Situation Awareness

Ten out of the fourteen respondents (71.4%) emphasized this theme during the interview. The participants indicated that visualizing vital parameters, particularly body temperature and oxygen saturation, helps them recognize underlying problems faster.


*“Even faster problem recognition, especially when the child’s temperature or oxygen saturation drops.”*
(Participant 2)

It was emphasized that the recognition of some parameters is better even from a considerable distance from the monitor. For example, during the induction of anesthesia for airway management, the anesthetists position themselves at the patient’s head while the monitor is often placed at the foot end of the patient, particularly in head surgery.


*“From the position on the patient’s head, e.g., during anesthesia induction (standing far from the monitor), changes in the vital parameters can be quickly recognized.”*
(Participant 3)

Participant 9 highlighted that the Visual Patient Avatar has the ability to quickly provide a generalized picture of the situation, leading to an improvement in situation awareness.


*“Visual Patient Avatar is good for situation awareness, e.g., if a lot of attention is taken up with the distraction of the child at the beginning of the induction, a good picture of the overall situation can be formed by a quick look at the Visual Patient Avatar. The focus is directly drawn to the problem if there is one.”*
(Participant 9)

#### 3.2.2. Usefulness of the Tool for Distracting Children and Reducing Stress

The second most frequently mentioned advantage—the effectiveness of the Visual Patient Avatar as a tool for distracting children from the procedures during induction and reducing perioperative stress—was emphasized by seven out of the fourteen respondents (50%).


*“Visual Patient Avatar offers the possibility of distracting children during certain procedures.”*
(Participant 2)

Respondents also emphasized that the Visual Patient Avatar has the ability to capture a child’s attention through its appearance and can assist in explaining the meaning of certain parameters, which adds an element of entertainment value.


*“Visual Patient is a good tool to entertain the child, exciting to look at, and attracts attention through flashing and colors. It is also good for explaining individual monitoring parameters to the child.”*
(Participant 9)

### 3.3. Areas for Improvement of Visual Patient Avatar from Anesthesia Providers’ Perspective

Two dominant themes emerged from the responses regarding areas for improvement of the Visual Patient Avatar in pediatric anesthesia: (1) the lack of adaptability of the avatar thresholds for children of different age groups and (2) the absence of quantitative information.

#### 3.3.1. Lack of Adaptability of the Visual Patient Avatar Thresholds for Children of Different Age Groups

Of the fourteen participants, seven (50%) felt that allowing the Visual Patient Avatar to adjust its thresholds according to different age groups would be beneficial for improvement.


*“There should be more profiles of Visual Patient Avatar for different age groups in pediatric anesthesia because with the standard pediatric profile, the alarms do not always fit with very small children or even older children, and it then alarms wrongly and distracts.”*
(Participant 9)

Specifically, regarding the heart rate, participants emphasized the necessity of adapting the Visual Patient thresholds for different age groups. It was observed that high pediatric heart rates, which exceed the preset threshold of the Visual Patient Avatar and cause its heart to pulsate rapidly, have the potential to distract attention.


*“The very high pediatric heart rate is visually distracting in Visual Patient Avatar.”*
(Participant 3)

#### 3.3.2. Lack of Quantitative Information

A second frequently mentioned theme regarding areas for improvement of the Visual Patient Avatar in pediatric anesthesia was the lack of quantitative information. This theme was mentioned by 5 out of the 14 participants (35.7%).

Participant 7 emphasized that, especially in critical situations, quantitative information is needed for decision-making:


*“In a critical situation, I would not pay much attention to visualizations. You need numbers for that.”*


### 3.4. Anesthesia Providers’ Comments on Children’s Reactions to Visual Patient Avatar

With the exception of one participant, who stated that children respond to the Visual Patient Avatar “neutrally,” all other participants (13/14, 92.9%) emphasized that children react positively and show interest in the Visual Patient Avatar.


*“Children react very positively to Visual Patient Avatar: curious, find it exciting or even funny.”*
(Participant 9)


*“Children like Visual Patient Avatar: its sympathetic appearance, friendly impression.”*
(Participant 1)


*“Children definitely benefit from Visual Patient Avatar: you can explain the process to them better, and it also has an entertainment value.”*
(Participant 6)

### 3.5. Anesthesia Providers’ Comments on Parents’ Reactions to Visual Patient Avatar

Five out of fourteen respondents (35.7%) mentioned that parents had expressed interest or a positive opinion about the Visual Patient Avatar.


*“I experience it in Caesarean sections: the patient and the partner react very positively to Visual Patient Avatar; so I imagine that parents in pediatric anesthesia settings would also respond positively.”*
(Participant 3)

Four out of fourteen participants (28.6%) expressed concerns about parents’ reactions to the Visual Patient Avatar, noting that it could potentially irritate parents if they are unaware of the meaning behind the changes in the Visual Patient Avatar.


*“Parents might be irritated, possibly worried, if everything flashes quickly or Visual Patient Avatar turns purple.”*
(Participant 9)

Figure 3 provides the participation rate and the main themes identified in the responses to the open-ended questions, with the number of participants and percentages.

## 4. Discussion

This study examined the observations and opinions of anesthesia personnel on the Philips Visual Patient Avatar in pediatric anesthesia after its real-life implementation in a single center. This study allowed us to identify the aspects of the technology that are positively regarded by pediatric anesthesia care providers and indirectly by patients and their parents or legal guardians. Moreover, it highlighted areas that could benefit from further improvements, according to their perspectives. Most notably, participants observed that the technology was well received by children during the perioperative phase and that it enhanced situation awareness. The most common suggestion for future improvement was the ability for users to adjust the visualization thresholds of the Visual Patient Avatar. A substantial 93% of participants emphasized the positive responses and interest that children demonstrated toward the Visual Patient Avatar. Around 50% of the participants also noted that the Visual Patient Avatar effectively distracts children during anesthesia induction, potentially helping to lower their perioperative stress levels. Even in cases of minor day surgery, substantial evidence indicates that children experience significant distress and anxiety during the perioperative period [31,32,33]. The anesthesia induction process can be a frightening experience for children, as it involves encountering an unfamiliar situation. Before surgery, as many as 75% of children undergoing anesthesia have reported experiencing anxiety or distress [34]. This anxiety is often manifested through behaviors such as crying, fear of separation, restlessness, anger, or behavioral freezing. There is no conclusive evidence on the effects of anesthesia in the early stages of a child’s development. More research is needed on the specific effects of general anesthesia and the effects of underlying health conditions that lead to the need for surgery or diagnostic procedures, especially in children who receive prolonged general anesthesia or have repeated hospitalizations [35,36,37,38]. So the fact that the Visual Patient Avatar, by distracting the child and arousing interest, may help to reduce anxiety in the perioperative period is a positive finding that may even influence the child’s further development positively.

The second most mentioned theme, reported by 71% of participants, was that the Visual Patient Avatar enhanced situation awareness. This finding demonstrates that the Visual Patient Avatar successfully achieves its primary goal as a situation awareness-oriented visualization technology. As shown in previous computer-based and high-fidelity simulation studies of the Visual Patient technology [4], integrated numerical and waveform data, represented in the visual form of changing colors, shapes and animations, contributed to faster decision making, increased accuracy of clinical diagnoses, decreased perceived workload, and increased perceived diagnostic confidence. These findings could be explained by the Visual Patient’s design, which integrates multiple parameters from multiple sensors into a single indicator, enabling care providers to assess multiple vital signs simultaneously [3,4]. These points are underscored by the principles of cognitive psychology and neuroscience that underpin data visualization. The Gestalt principles of perception—similarity, proximity, closure, enclosure, continuity, and connection—suggest that when humans encounter visual stimuli, they do not just passively observe individual elements. Instead, the human mind actively organizes and interprets these elements, perceiving them as unified and meaningful entities [7,8].

An idea for improvement identified in this study was the possibility of providing more profiles for more different age groups of pediatric patients. Half of the participants mentioned this suggestion as a valuable improvement that could enhance their practice.

### Limitations and Strengths

This study has several limitations and strengths. First, the study is subject to the inherent limitations of qualitative research. Findings derived from qualitative analysis cannot be generalized to larger populations with the same level of certainty as quantitative results, as they are not tested for statistical significance [39]. Moreover, there is a lack of triangulation due to the novelty of the technology and the fact that the study center is a pioneer center for Philips Visual Patient Avatar, and the current study is the first to examine the technology after its real-world implementation. Furthermore, it is important to note that this was a relatively small study conducted at a single university hospital in Europe, the initial development site for the technology, with a specialized user group of pediatric anesthesiologists. As a result, user perceptions may differ across various clinical settings worldwide, and additional qualitative and quantitative studies are needed. Furthermore, participant selection for the interviews relied on their availability within their clinical routine and their willingness to participate in an interview on this topic, as opposed to a randomized selection. Study participants work at the center where the technology was developed, which may have biased their responses. Finally, despite our efforts to address these concerns, our survey results may be affected by nonresponse [27], self-selection [28], and information bias [29]. Nonresponse can result in incomplete and potentially biased data if certain individuals choose not to participate. As a result, the findings may not accurately reflect the broader population, which may affect the generalizability of the results. In addition, if individuals choose to participate based on their personal experiences, interests, or motivations, the sample may not adequately represent the entire target population. While we have taken steps to create a robust data collection instrument to maximize generalizability, information bias remains a factor that can influence survey results. It is important to note that our study included clinicians with demanding clinical responsibilities, which may have contributed to nonparticipation due to time constraints. There is a need for larger, multi-center studies with advanced designs to fully capture users’ perceptions of the new technology. Future research should prioritize diverse demographics and contexts and use mixed-methods approaches and usability scores for a nuanced understanding.

A significant strength of this study lies in its novelty; it is the first study to investigate the perceptions of users and, indirectly, pediatric patients and their parents or legal guardians, regarding the Visual Patient Avatar technology after its implementation in clinical practice.

## 5. Conclusions

This study is the initial exploration into the perceptions of pediatric anesthesia providers, patients, and their parents or legal guardians regarding the Visual Patient Avatar technology after its integration into clinical practice. The findings of the study indicate that pediatric anesthesia providers perceive the visual information provided by the Visual Patient Avatar as a valuable tool for rapidly identifying underlying issues and, consequently, improving situation awareness. A notable idea for improvement proposed by the participants was to offer optional adjustments to the Visual Patient thresholds based on pediatric age groups. An essential piece of feedback was the remarkable appreciation expressed by the patients—the children. Not only does this technology show potential in improving the technical aspects of anesthesia, but it may also offer the promise of alleviating perioperative stress and enhancing the psychological acceptability of these procedures for children. This added value further underscores the significance of the technology.

## Figures and Tables

**Figure 1 children-10-01841-f001:**
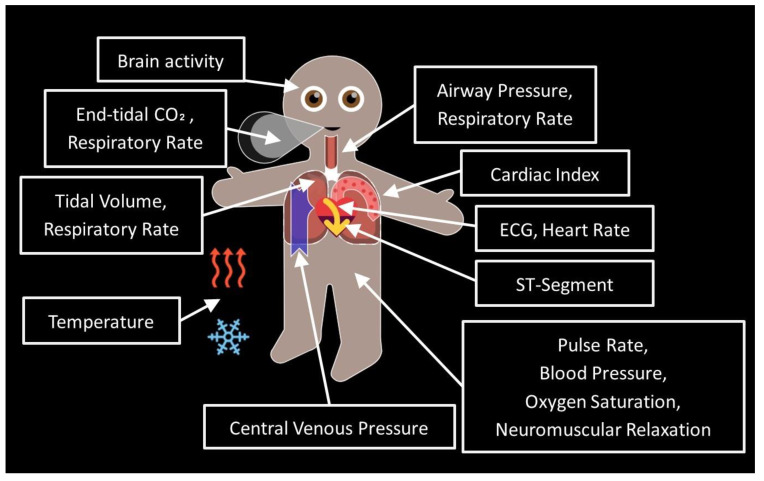
Visual Patient Avatar: Vital signs and locations of their appearance.

**Figure 2 children-10-01841-f002:**
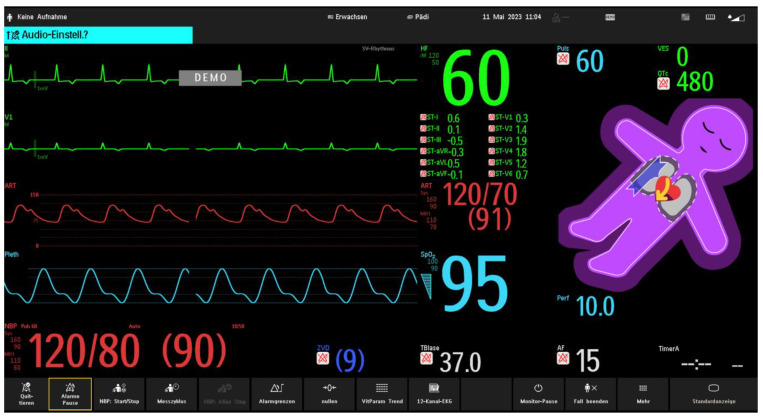
An example of a split–screen view, providing a combination of conventional monitoring and the Visual Patient Avatar. The split screen shows real-time data from traditional patient monitoring parameters such as heart rate (green on conventional monitoring), blood pressure (red on conventional monitoring), and oxygen saturation (blue on conventional monitoring), alongside the Visual Patient Avatar. This allows healthcare providers to have a comprehensive and integrated view of the avatar–based visualization and the conventional monitoring information simultaneously. The purple color of Philips Visual Patient Avatar indicates that the oxygen saturation is below the set threshold.

**Figure 3 children-10-01841-f003:**
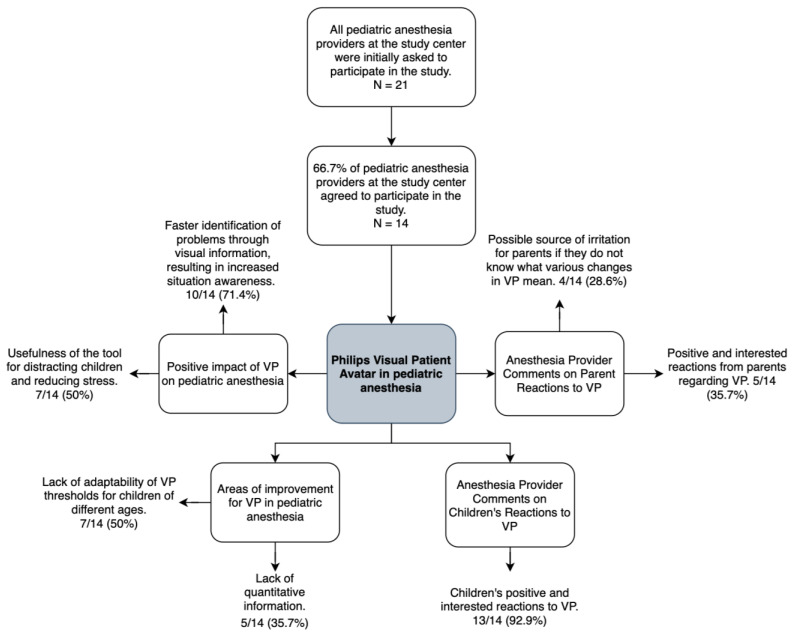
Flowchart illustrating the participation rate and the main themes identified in the responses, with the number of participants and corresponding percentages. Abbreviations: VP, Philips Visual Patient Avatar.

**Table 1 children-10-01841-t001:** Participant characteristics.

Participants (*n* = 14)	
Sex	
Female	5 (35.7%)
Male	9 (64.3%)
Participant age in years, median (IQR)	42 (38–45)
Anesthesia experience in years, median (IQR)	11 (6–14)
Role	
Certified nurse anesthetist	4 (28.6%)
Resident 1–2 years of training (limits: 1 year to 2 years)	1 (7.1%)
Resident 3–5 years of training (limits: 3 to 5 years)	1 (7.1%)
Resident with >5 years of experience (limits: 6 years to end of residency)	3 (21.4%)
Staff anesthesiologist	5 (35.7%)
Educational material consulted	
Quick guide only	4 (28.6%)
Tutorial video only	0
Both	3 (21.4%)
None	7 (50.0%)

Abbreviations: IQR, interquartile range.

## Data Availability

The data presented in this study are available in the Appendix A.

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
