# Peer review of "Pediatric Anesthesia Providers’ Perspective on the Real-Life Implementation of the Philips Visual Patient Avatar: A Qualitative Study"

_children, 2023, doi:10.3390/children10121841_

Round 1
Reviewer 1 Report
Comments and Suggestions for Authors
An interesting study is presented that evaluated pediatric anesthesia staff's perspective on the routine use of a visual patient avatar.
The introduction is well-founded, those studies that favor the use of this type of tool are presented. In this regard, we would like to know if you found any article that does not recommend this type of tool and what would be the reason?
The methodology talks about the use of an interview guide. Was this interview guide standardized and validated for this purpose by expert personnel? Could you present the results of this activity?
Were possible sources of bias identified?
Explain how the sample size was determined.
Indicate the statistical methods used for numerical variables, qualities and to control confounding factors.
Add a flowchart.
If you classify continuous variables, describe the limits of the intervals.
If appropriate, consider accompanying relative risk estimates with absolute risk estimates for a relevant time period.
Reviewer 2 Report
Comments and Suggestions for Authors
This manuscript describes a novel tool, the Philips Visual Patient Avatar, and describes its use in pediatric anesthesiology. As a supplement to the traditional anesthetic monitoring tools, this device utilizes colors, animations, and shapes to convey information. This study was designed as a qualitative analysis and provides providers' perceptions of this new and unique monitoring technology.
Although this paper presents a novel tool that may be of interest to the reader, the overall study design is poorly constructed. Residents and nurses who agreed to be a part of the study provided commentary on their perception of the tools usefulness and areas of improvement. As this product was developed at their institution by members of the staff, they may have inherent bias or felt obligated to give favorable reviews. The results described were descriptive based instead of a usability score or rating. Further, a study design were providers were monitored for how quickly they responded to a variation based on the avatar vs traditional monitoring would provide the best design. Therefore, I think this manuscript provides minimal clarity on the utilization of this device.
Specific comments:
Abstract: Line 22-23. The statement that the Avatar was recognized as a valuable tool is suggestive and is not warranted by the results. While 71% reported faster recognition of vital sign derangement, 50% reported that the device could not be set by age group and altered them to variations that were normal.
Introduction: line 46. The authors decisive situational awareness as comprehending the significance of an issue. While this avatar may increase situational awareness in that it has flashing lights and animations that may grab the provider's attention better but it does not mean that a provider better understands the significance of that change.
Figure 1: The figure provides minimal description of how the avatar changes based on vital derangements. Instead, it should have a graph with all avatar settings.
Materials and Methods:
Line 81-82. It is not appropriate to utilize residents and nurses who work with the inventors of the avatar as it can clearly bias their opinion.
Line 88. How long were providers exposed to this technology before evaluating it? This is important to clarify if they had 50 exposures versus 1 exposure as this could change their perception of its use.
Line 110: A usability score should have been performed by each participant. The use of an open-ended question format provides no clear discernible data and lends itself to ambiguity.
Results: Table 2. This table is poorly formatted as it does not give a quick reference to perceptions.
Discussion:
Line 277. These papers are from 2004 and there are newer studies that are contradictory to this statement. I would include references to the more recent studies on anesthesia and neurodevelopment behavior outcomes.
Line 282: As this was purely a qualitative study, there is little data to support that the avatar is successfully achieving increased situational awareness. This would need actual observational data on providers response to avatar changes.
Line 309: The authors do not mention in limitations that these trainees and nurses may potentially be biased as they work at the hospital were the avatar was designed.
Reviewer 3 Report
Comments and Suggestions for Authors
1. Brief Summary and General Comments
- First, I would like to congratulate the authors on their interesting qualitative study about paediatric anaesthesia providers’ perspectives on a visual patient avatar. The authors interviewed 14 members of the local Institution’s paediatric anaesthesia team and present themes identified along with the respective interview fragments. I recognized the following key messages in this manuscript:
o The visual patient avatar can support medical issue identification, thereby improving the situation awareness of anaesthesiology staff in the OR.
o It may be useful to distract children from the current situation in the OR which may reduce their stress level. All in all, anaesthesia providers rated parents’ and children’s perceptions of the visual patient avatar as being positive.
o There is a need for further improvement of the avatar (e.g., by adding thresholds for different age groups and quantitative information).
Given that your study design is a qualitative study, I highly recommend considering the quality criteria presented in the literature for qualitative studies (Frambach et al., 2013; Tracy, 2010). In addition, the following comments could be considered during revision:
2. Abstract
- Please delete lines 26-27 as it is the template for adding keywords to the manuscript.
3. Introduction
- Given the qualitative research design, I would kindly advise you to add the research questions to the purpose of the study.
- What was your approach to inquiry (descriptive, interpretive, constructivist…)?
4. Methods
- Could you please describe the selection process of the participants and the in-/exclusion criteria applied?
- Figure 2: Please consider adding the colour code for the parameters as presented in your figure to your legend.
- Please add a reflexivity paragraph to this section. The AMEE guide no 149 published by Olmos-Vega and colleagues could give you some helpful insights (Olmos-Vega et al., 2023).
- Please describe the data analysis techniques used in more detail. Which attempts were made to assess the rigor of your study (e.g., internal validity, external validity, reliability)?
5. Results
- Table 1: Please add the abbreviations used to the legend.
6. Discussion
- Please consider discussing the lack of triangulation as limitation of your study.
- Please add implications for future research on this topic to the final paragraph.
Thank you in advance for considering the comments during revision. Good luck to the authors and kind regards!
References
Frambach, J. M., van der Vleuten, C. P. M., & Durning, S. J. (2013). AM last page. Quality criteria in qualitative and quantitative research. Academic Medicine: Journal of the Association of American Medical Colleges, 88(4), 552. https://doi.org/10.1097/ACM.0b013e31828abf7f
Olmos-Vega, F. M., Stalmeijer, R. E., Varpio, L., & Kahlke, R. (2023). A practical guide to reflexivity in qualitative research: AMEE Guide No. 149. Medical Teacher, 45(3), 241–251. https://doi.org/10.1080/0142159X.2022.2057287
Tracy, S. J. (2010). Qualitative Quality: Eight “Big-Tent” Criteria for Excellent Qualitative Research. Qualitative Inquiry, 16(10), 837–851. https://doi.org/10.1177/1077800410383121
Comments on the Quality of English LanguageOverall, the manuscript could benefit from minor editing of English language.
Round 2
Reviewer 1 Report
Comments and Suggestions for Authors
OK
Reviewer 2 Report
Comments and Suggestions for Authors
This manuscript provides a unique perspective on a novel avatar tool in pediatric anesthesia. Thank you to the authors for their thoughtful considerations of the comments and edits to the manuscript. The new flowsheet and updated figures provide a much more comprehensive understanding of the project. Overall, the manuscript is well written and provides an appropriate summary of the information obtained.
Reviewer 3 Report
Comments and Suggestions for Authors
1. General Comments
Dear authors,
Thank you for considering my comments during the revision of your manuscript. I think your efforts made have improved the manuscript significantly. In the current version of your manuscript, I have identified a few further comments which could be considered during revision:
2. Introduction
- Line 60-61: Please add a reference to this sentence.
- Could you please elaborate a bit more on the cognitive analyses of specific tasks by Sowb & Loeb and Roth mentioned in line 68-70?
3. Methods
- Line 137-139: How could your study have been influenced by your personal reflexivity (e.g., your motivations, experiences, expectations)?
4. Results
- Figure 3: Please explain the abbreviation VP used in your figure in the legend.
Thanks again for considering my comments during revision and all the best wishes!
Comments on the Quality of English LanguageThe manuscript could benefit from some minor editing of English language.
